# Impact of Cultivation Substrate and Microbial Community on Improving Mushroom Productivity: A Review

**DOI:** 10.3390/biology11040569

**Published:** 2022-04-08

**Authors:** Nakarin Suwannarach, Jaturong Kumla, Yan Zhao, Pattana Kakumyan

**Affiliations:** 1Research Center of Microbial Diversity and Sustainable Utilization, Faculty of Science, Chiang Mai University, Chiang Mai 50200, Thailand; suwan.462@gmail.com (N.S.); jaturong_yai@hotmail.com (J.K.); 2Institute of Edible Fungi, Shanghai Academy of Agricultural Sciences, Shanghai 201403, China; 3School of Science, Mae Fah Luang University, Chiang Rai 57100, Thailand

**Keywords:** mushroom cultivation, lignocellulosic substrate, solid fermentation, microbial community

## Abstract

**Simple Summary:**

Lignocellulosic material and substrate formulations affect mushroom productivity. The microbial community in cultivation substrates affects the quality of the substrates and the efficiency of mushroom production. The elucidation of the key microbes and their biochemical function can serve as a useful guide in the development of a more effective system for mushroom cultivation.

**Abstract:**

Lignocellulosic materials commonly serve as base substrates for mushroom production. Cellulose, hemicellulose, and lignin are the major components of lignocellulose materials. The composition of these components depends upon the plant species. Currently, composted and non-composted lignocellulosic materials are used as substrates in mushroom cultivation depending on the mushroom species. Different substrate compositions can directly affect the quality and quantity of mushroom production yields. Consequently, the microbial dynamics and communities of the composting substrates can significantly affect mushroom production. Therefore, changes in both substrate composition and microbial diversity during the cultivation process can impact the production of high-quality substrates and result in a high degree of biological efficiency. A brief review of the current findings on substrate composition and microbial diversity for mushroom cultivation is provided in this paper. We also summarize the advantages and disadvantages of various methods of mushroom cultivation by analyzing the microbial diversity of the composting substrates during mushroom cultivation. The resulting information will serve as a useful guide for future researchers in their attempts to increase mushroom productivity through the selection of suitable substrate compositions and their relation to the microbial community.

## 1. Introduction

Mushrooms are an acknowledged food source and are widely consumed throughout the world. Global mushroom cultivation was estimated at approximately 11.9 million tons per year in 2019 [1]. China is the largest mushroom producer (8.9 million tons), followed by Japan (0.47 million tons) and the United States of America (0.38 million tons). Several species of edible mushrooms belonging to the phylum Basidiomycota, in the genera *Agaricus*, *Agrocybe*, *Auricularia*, *Flammulina*, *Ganoderma*, *Hericium*, *Lentinula*, *Lentinus*, *Pleurotus*, *Tremella*, and *Volvariella,* have been commercially cultivated [2,3,4]. *Agaricus bisporus* (button mushroom), *Pleurotus* species (oyster mushroom), *Lentinula edodes* (shiitake), *Auricularia* species (wood ear mushroom), *Flammulina velutipes* (enoki), and *Volvariella* species (paddy straw mushroom) are the most commonly cultivated mushrooms in the world [3,4,5]. The increasing trend of mushroom production is expected to continue in the future. Mushrooms are heterotrophic organisms that require external nutrients to grow vegetative mycelium and reach the reproductive stage (fructification) [2,6]. Therefore, most cultivated mushrooms are saprophytic fungi or decomposers. These can grow on lignocellulosic substrates by producing several lignocellulosic enzymes to degrade substrates as they grow [2,7]. Solid-stage cultivation under controlled conditions is a common method used for commercial mushroom production. However, the substrate composition and formula for mushroom cultivation need to be optimized, depending on the species. Some cultivated mushroom species, e.g., *Auricularia*, *Lentinula*, and *Pleurotus*, can be cultivated under “axenic conditions” using nondegraded sterile substrates. Accordingly, their fruiting bodies are directly produced from a lignocellulosic substrate. In contrast, some mushroom species, e.g., *Agaricus* species, are classified as secondary decomposer fungi that specifically grow on partially decomposed, humic-rich substrates [8,9]. Therefore, the substrate for the cultivation of the *Agaricus* species must be prepared through aerobic solid-state fermentation (composting) in order to provide the necessary available carbon and nitrogen sources with humic-rich complexes. Soil organic matter, or the casing layer, is required to stimulate fruitbody formation for some mushrooms, e.g., the *Agaricus* species [10] and the *Morchella* species [11]. Nutritional supplements, such as rice bran, animal manure, grain, gypsum, limestone, and synthetic additives, are sometimes added to the principle material. It has been speculated that the casing materials contain bacteria involved in the transformation of vegetative mycelial growth into fruiting body formation [9]. It has recently been reported that different substrate materials lead to different mushroom production efficiencies. More and more studies are focusing on the effect of the microbial community on substrate quality during substrate preparation. However, an in-depth understanding of the mechanisms involved in productivity has yet to be elucidated. Understanding the key microbes involved in each process of the mushroom cultivation cycle will be of significant use for the improvement of mushroom cultivation in the future, especially for those species growing on substrates produced through aerobic solid-state fermentation.

Understanding cohabitating microorganisms in mushrooms could significantly benefit mushroom science, and support the development of strategies that would enable researchers to increase both productivity and mushroom quality. Although detailed studies on the relations between edible fungal growth and the microbial community in substrates have been widely reported, there are still gaps in the existing knowledge due to the involvement of various heterogeneous raw materials and mushroom species. Molecular techniques have helped researchers study the entire microbial community of cultured and non-cultured species during mushroom cultivation, rather than solely focusing on the cultured species. To date, little information can be found on some mushroom species with regard to changes in the physicochemical properties of the fermentation media, the microbial community, and the metabolic functions involved in the composting and cultivation processes. Therefore, numerous composting and cultivation systems have been investigated. This has resulted in confusion among researchers in understanding these relationships. Multiple research teams have investigated the effect of the microbial community on the preparation of cultivation substrates or mushroom growth. Most of these studies have been performed on the *Agaricus* species during the casing and the composting processes [12,13]. Some of these studies have involved *Pleurotus* spp. [14,15], *Cordyceps militaris* [16], and *Ophiocordyceps highlandensis* [17]. Nevertheless, little information on the microbial dynamics during substrate preparation or the cultivation of other mushrooms is known. Therefore, the characterization of microbial communities could help researchers fully understand the functional structure of the community during substrate preparation and mushroom cultivation, in terms of mushroom growth and biological efficiency.

This review aimed to identify certain relevant and important factors, such as the composition of cultivation substrates and microbial communities, involved in mushroom cultivation. The methods used to analyze the microbial community in the cultivation process are summarized, and the advantages and disadvantages of each analysis method are reviewed. The resulting knowledge could be used by researchers to overcome the limitations of mushroom cultivation through microbial consortia in substrate preparation systems and mushroom growth-promoting microorganisms.

## 2. Mushroom Cultivation

### 2.1. Sources and Composition of Substrates

Most agro-industrial waste is generally burned or disposed of in landfills. However, these methods can cause environmental pollution that poses potential harm to the health of humans and wildlife [18,19]. Using agricultural materials as substrates for mushroom production is an alternative method used to reduce the negative impact of their disposal on the environment. Certain types of agricultural waste can serve as substrates for mushroom cultivation due to their high availability and cost efficiency. The authors of several studies have already employed agricultural waste for mushroom cultivation. Waste produced by the agriculture, forestry, horticulture, textile, and wood industries contains lignocellulosic substances that can be used to grow mushrooms [20]. Two types of agro-industrial waste, agricultural field residue and process residue, are generated from certain agricultural activities. Leaves, roots, stalks, straw, seed pods, and stems are classified as agricultural field residues that are generated during the crop-harvesting process. Husks, peels, pulp, bran, pomace, bagasse, and shells are classified as agricultural process residues that are generated during the processing of crops [4]. Types of agro-industrial waste that have been used for mushroom cultivation are summarized in Table 1.

Mushrooms can grow on numerous types of substrates, but the rates of substrate utilization and mushroom growth are dependent on the actual mushroom species [4]. Most agricultural waste contains lignocellulosic biomass as a major component. The composition of cellulose, hemicellulose, and lignin in lignocellulose materials depends on the plant species. This is an important factor that can impact mushroom cultivation. The substrates used for the effective cultivation of mushrooms are shown in Figure 1. *Volvariella volvacea* requires substrates with high cellulose and low lignin contents; it produces a variety of cellulase enzymes for cellulose degradation. These enzymes enable the mushrooms to utilize the substrate [50]. Rice straw is widely used as a substrate for *V*. *volvacea* strains, and this substrate is available in many countries. Alam et al. [51] reported that the maximum biological efficiency (BE) of *V*. *volvacea* could be obtained on the combined media of rice straw and wheat bran. The main substrates for the cultivation of *Pleurotus* spp. are hardwood sawdust, paddy straw, wheat straw, corn cobs, and cottonseed hulls [52]. Wheat straw is widely used as the base substrate in the cultivation of the *Agaricus* species.

A number of factors are required for the appropriate and efficient production of mushrooms. The factors pertaining to oxygen and moisture content, carbon and nitrogen ratio (C/N ratio), and optimal pH and temperature must be considered for successful cultivation. Macronutrients (C, N, P, K and Mg) are required by mushrooms, and certain trace elements, such as Fe, Se, Zn, Mn, Cu, and Mo, are needed for a range of diverse functions [2,53]. Reductions in temperature and the amount of CO_2_ exerted are also essential for the differentiation of primordia [3]. Some mushroom species require changes in pH or light conditions for the formation of primordia and fruiting bodies [3,53]. Bellettini et al. [54] described factors that affect the production of *Pleurotus* sp., including substrate composition, nitrogen source, C/N ratio, pH, moisture, minerals, material particle size, spawning level, surfactants, temperature, air composition, and humidity.

Different substrate formulations of mushroom cultivation can affect the biological properties of the substrate (population and activity of microbes) and the biological efficiency of the mushroom itself [4,55,56]. There are two main substrates used for commercial mushroom cultivation from agro-industrial waste: composted substrates (fermentation substrates) and non-composted substrates. Kong et al. [15] summarized changes in the physicochemical properties of compost leading to different genera at different composting stages in corncob composting systems. The use of different formula substrates may lead to different communities of microorganisms. Lignocellulosic substrates are degraded and yield a number of principal products including cellulose, hemicellulose, and lignin. Cellulose and hemicellulose are carbohydrates that act as carbon sources. Lignin provides carbon that is used by mycelium. Ultimately, lignin is converted into a nitrogen-rich lignin–humus complex [57,58]. The composting or fermentation process is a particularly important process involved in altering the biological and physiochemical properties of the growing substrate.

Mixtures of various agricultural materials can be used in mushroom cultivation [20]. Sugarcane bagasse, corncobs, soya stalks, sunflower stalks, ramie stalks, kenaf stalks, bulrush stalks, banana leaves, cotton seed hulls, bracts of pineapple crowns, oil palm fronds, and oil palm empty fruit bunches are examples of different types of agricultural residue that can supplement or replace the main substrates (rice straw and sawdust) (Table 1). These agricultural residues consist of lignocellulosic components that vary depending on the species of the plant. The C/N ratio is an important factor in mushroom production. To obtain profitable mushroom yields, the C/N ratio of the compost or substrate must be controlled. Cellulose is the most abundant component, followed by hemicellulose and lignin. However, most agricultural residues are defined as materials with a low nitrogen content. Therefore, the supplementation of organic nitrogen (bran, cereal shell, manure, and soybean meal) or inorganic nitrogen (ammonium chloride and urea) into substrates for mushroom cultivation is necessary [59,60]. Different mushroom species require different C/N ratios in the cultivation substrate. For example, the optimum C/N ratio for *A. bisporus* and *A. bitorquis* was found to be 19/1 [61,62,63], but *A*. *subrufescens* requires a C/N ratio of 27/1 [63,64]. The optimum C/N ratios for *L. edodes, G. lucidum,* and *V. volvacea* have been reported as 30–35/1, 70–80/1, and 40–60/1, respectively [65,66,67]. The authors of several studies on the cultivation of *Pleurotus* species have reported C/N ratios ranging from 45 to 60/1, depending on species [68,69,70,71,72]. Moreover, the addition of Epsom salts, gypsum, and limestone to the substrates also supports the mycelia growth and fruiting body production of mushrooms [71,73,74].

### 2.2. Methods for Mushroom Cultivation

Edible mushrooms can be classified as primary, secondary, and tertiary decomposers. Primary decomposers, such as oyster and shiitake mushrooms, can directly grow on plant materials. Secondary decomposers, such as button and straw mushrooms, typically grow on fermented or composted materials. *Volvariella volvacea* and *A*. *bisporus* play a significant role in the degradation of straw, but a diminished role in degrading the lignin component of lignocelluloses [75,76]. Tertiary decomposers, such as *Agrocybe* spp., are generally present in soil [77]. Successful mushroom cultivation requires both scientific knowledge and practical experience. Three specific methods can be used for the effective cultivation of mushroom species. These include using raw materials without sterilization or composting, autoclaving to accomplish sterilization, and certain composting cultivation methods [15,78,79,80]. The cultivation methods used in mushroom production are shown in Figure 2. Some mushrooms require casing soil, and will form few or no primordia without it. *Agaricus bisporus* is an example; it requires a casing overlay to cover the colonized substrate to induce mushroom fructification [44,81].

#### 2.2.1. Cultivation on Non-Composted Substrates

Non-composted substrates are used in the cultivation of a number of mushrooms, including *Agrocybe cylindracea*, *Auricularia* species, *Flammulina velutipes*, *Ganoderma lucidum*, *Hericium erinaceus*, *Lentinula edodes*, *Lentinus sajor*-*caju*, *Pleurotus* species, and *V. volvacea*, as shown in Table 1. The substrates can be prepared, sterilized, or pasteurized, and then used in their natural form. Various types of cultivation substrates, including cottonseed hulls, bulrush stalks, ramie stalks, and kenaf stalks, were used to study their effects on lignocellulosic biomass degradation, lignocellulosic enzyme production, and biological efficiency in *P*. *eryngii* by Xie et al. [33]. The results indicate that the yields and BE values obtained from different substrate cultivations significantly varied (from 36.8 to 52.4%). The kenaf stalks were found to the most suitable for cultivation, with a BE value of 52.4%. Sardar et al. [34] also confirmed that this mushroom was able to grow on various substrates, including wheat straw, rice straw, sawdust, corn cobs, and sugarcane bagasse, with BE values ranging between 35.47% and 71.56% depending on the substrate. The contents of cellulose, hemicellulose, and lignin decreased after fungal growth, and lignocellulosic degradation efficiency had a positive correlation with the BE value. The percentage loss of lignin in the wheat straw substrate during *P*. *ostreatus* production was higher than that in the combined wheat straw with olive pruning residues or spent coffee grounds (53.51%, 26.25%, and 46.15%, respectively), reflecting better mycelia growth and leading to the slightly higher BE values in the wheat straw substrate than in the combined wheat straw with olive pruning residues or spent coffee grounds (105.0%, 95.3%, and 101.7%, respectively) [40]. Notably, the biological efficiency of *H*. *erinaceus* was positively correlated with lignin content and inversely correlated with the ratio of cellulose and lignin. Different kinds of sawdust affected the yield and BE value of *H*. *erinaceus* [22]. *Ganoderma lucidum* was able to grow on oat straw and sawdust substrates. However, the type of sawdust tree used affected the BE value [27]. Sawdust is an agricultural substrate that can be used for *A*. *polytricha* cultivation, and supplementation with substrates, such as oil palm fronds and empty fruit bunches, to sawdust substrates could enhance BE values [28,29]. Various types of substrates, such as rice straw, wheat straw, barley straw, sugarcane leaves, and bagasse, have been used to study their efficiency for *L*. *edodes* production. Remarkably, the type of straw has been shown to have an effect on mushroom yields. Accordingly, Gaitán-Hernández et al. [31] found that the BE value of *L*. *edodes* in a barley straw substrate was higher than that in wheat and rice straw substrates. Sugarcane bagasse and leaves have the potential to be used as substrates to effectively grow *L*. *edodes*, as high BE values were observed and reported by Salmones et al. [32]. Wheat straw or sawdust supplemented with wheat bran is commonly used as a substrate for *A*. *cylindracea* cultivation. The yield of *A*. *cylindracea* grown on wheat straw was found to be higher than that grown on beech sawdust [36]. Numerous agricultural materials can be used for mushroom cultivation, and a mix of materials can sometimes enhance mushroom productivity.

#### 2.2.2. Cultivation on Composted Substrates

The cultivation of mushrooms using non-composted and composted substrates has been investigated, and the degrees of efficiency have been compared in order to identify any alternative cultivation methods that could reduce the cost and time associated with production. Button mushrooms (*Agaricus* species) and paddy straw mushrooms (*Volvariella* species) are commonly cultivated with the use of composting substrates. The standard process of substrate preparation for these methods of mushroom cultivation involves composting. Two phases, composting phases I and II, of the composting process have been identified [10,82]. Mesophilic organisms utilize easily accessible nutrients during the water-wetting process at the beginning of phase I. Phase I is a thermobiological fermentation process that involves the bioconversion of raw materials and the release of nutrients. In this process, the temperature in the system rises to 80 °C. Phase II begins with pasteurization and continues with the conditioning of volatile ammonia from the production process. The substrate is ready for use in mushroom production following the removal of the ammonia from the compost. The biomass in the substrates is degraded, and releases nutrients during the fermentation process due to the microbial communities in the compost that produce the enzymes involved in biomass deconstruction. Promoting synergistic microbial communities is discussed in the next section on microbial communities for mushroom cultivation. The formulation of compost is an important variable in the productivity of mushroom cultivation. Wheat straw is normally used for the production of *Agaricus* spp., whereas rice straw and cotton seed waste are normally used for *Volvariella* spp. The basic components of compost are straw, animal manure, and gypsum. Supplemental ingredients, such as bran, urea, calcium, and ammonium salts, are required depending on the species of mushrooms. Additionally, *Pleurotus* species can grow on various kinds of substrates, though BE values were found to improve by over 50% on composted sawdust and rice straw substrates. Obodai et al. [37] compared the effects of different substrates and the composting of those substrates on the growth and yield of *P*. *ostreatus*. It was found that composted sawdust yielded higher BE values (61%) than non-composted sawdust (4.3%). Furthermore, the composting of the substrate tends to lead to higher yields of *P*. *ostreatus* and is necessary for *Agaricus* species. Therefore, the processes of composting must be adjusted based on the types of lignocellulosic materials being used and the initial substrate composition.

### 2.3. Enzymes Involved in Substrate Utilization

Most substrates used for mushroom cultivation are complex structures that are mainly composed of cellulose, hemicellulose, and lignin. A set of hydrolytic enzymes is needed for substrate degradation and saccharification in order to fully utilize all of the substrate components. Carbohydrate-active enzymes (CAZymes), including glycosyltransferase, glycoside hydrolases, carbohydrate esterases, polysaccharide lyases, and carbohydrate-binding modules, involve the breakdown, biosynthesis, or modification of glycoconjugates, as well as oligo- and polysaccharides such as cellulose, hemicellulose, and pectin [11,83]. Oxidoreductases consist of auxiliary activities (AAs) and cytochrome P450 enzymes. Oxidoreductases are associated with lignocellulose conversion, which involves lignocellulolytic enzymes, such as cellulase (endo-β-1,4-glucanase, exo-β-1,4-glucanase, and β-glucosidase), hemicellulose (endo-β-1,4-xylanase, β-xylosidase, α-L-arabinofuranosidase, and mannanase), and lignin-degrading enzymes (lignin peroxidase, manganese peroxidase, laccase, and versatile peroxidase), which then become involved in the conversion of lignocellulosic biomass into soluble sugars [33,76]. Cytochrome P450 monooxygenases are mixed functional oxidoreductases, and play an important role in the biosynthesis of numerous secondary metabolites downstream of the lignin-degradation process [76,84,85].

Many of the enzymes involved in biomass material degradation have been previously identified and described. Enzyme production profiles are affected by the type of substrate, growth phase, and species of mushrooms. For example, cellulase, hemicellulase, and lignin depolymerization enzymes are significantly produced by *P*. *eryngii* depending on the type of lignocellulosic substrate. The cellulose, hemicellulose, and lignin contents in the substrates of *P*. *eryngii* cultivation are decreased during the process of cultivation, in which the efficiency of lignocellulosic biomass degradation was found to have a positive correlation with biological efficiency [33]. *Pleurotus ostreatus* grown on cotton waste substrate was found to significantly produce cellulase and xylanase during primordia and fruiting body development, and laccase and manganese peroxidase were found to be highly produced during the colonization stage, and to decline during the primordia and fruiting body formation stages [86]. Xing et al. [87] observed laccase production in the cultivation of *Grifola frondose* (maitake) on beech sawdust, as well as that laccase activity increased during the colonization phase and declined during fruiting body development. In contrast, low levels of laccase production in the cultivation of *V*. *volvacea* on cotton waste compost were detected during the vegetative growth phase, but the enzyme concentration significantly increased during fruiting body development [88]. Zhang et al. [89] found that the endo-xylanase and mannanase peroxidase productivities of *A*. *bisporus* cultivated on straw compost were increased during the development of pinning to the fruiting body, but these enzymes decreased during mature fruiting body development. Endo-cellulase or endo-β-1,4-glucanase were found to increase during days 15–45 (colonization stage) of the cultivation of *Morchella importuna,* and peaked during days 45–75 (fruiting body formation stage). Remarkably, the amounts of exo-cellulase or exo-β-1,4-glucanase and β-glycosidase were much lower than that of endo-cellulase [11]. β-glucosidase was highly produced during days 20–30 (colonization stage) of the cultivation of *G. frondose,* and the activity was constant up to day 75 of cultivation (primordia formation). Substrate composition can also affect enzyme production [90]. Besides the lignocellulosic enzymes produced from mushrooms, some microbes in the substrate fermentation system in composted substrates also produce the enzymes needed to degrade the substrate, and provide soluble sugars that support mushroom growth [91,92].

## 3. Microbial Community for Mushroom Cultivation

### 3.1. Methods Used for Analysis of Microbial Communities

To emphasize how the biodiversity of microbes or community compositions affects cultivation or environmental changes, more knowledge regarding the microbial community, including both the bacterial and fungal communities in composted or cultivated substrates, needs to be acquired. The microbial communities associated with mushroom growth have been studied with culture-dependent and culture-independent methods. Culture-independent methods have been used to reveal the immense diversity of uncultured organisms, and to implement complementary approaches for the analysis of microbial diversity in entire systems [93]. However, not all microbial diversity in the system is accessible with culture methods because of the limitations of traditional enrichment methods and pure culture techniques.

#### 3.1.1. Culture-Dependent Methods

The diversity of microorganisms is typically identified based on the morphology, physiology, and phylogenetics of microorganisms after isolation and cultivation. The metabolic characteristics of carbon sources can then be used to evaluate the diversity of microbial communities.

##### Plate Culture Method

An appropriate medium according to the target microorganism is used in the plate culture method to determine the total number of different microbial communities present. The species of the cultured organisms are then identified according to applicable physiological, chemical, and morphological characteristics. This method provides relevant information on the active and heterotrophic components of microorganisms. Salar and Aneja [94] found that thermophilic fungi isolated using the plate culture method were prevalent in the fermentation materials of *A*. *bisporus,* and that most of these fungi could promote the growth of the hyphae. Notably, the culture conditions and the actual living environment of microbes are not exactly the same, and only some of the microbes can be cultured and further isolated in a petri dish through this method. Importantly, the cultivation conditions of most microorganisms are strict and difficult to replicate. Accordingly, this method can only detect the presence and function of some microbes in fermented materials, though it can be used to obtain pure cultures of the microorganisms [95].

##### Biolog Microplate Method

The Biolog microplate method is based on the study of the metabolic diversity of microorganisms used to oxidize carbon sources. This method reflects the overall activity of the microbial community, and characterizes microbial functional diversity without the need to analyze individual microorganisms. In this method, microbial communities are distinguished by analyzing the color of the dye and the number of micropore reactions on the micropore plate. This method has been widely used to characterize the physiological characteristics of microbial communities in different substrates [96,97,98]. However, the application of biological microplate methods in the research of fermentation materials is relatively limited. Farnet et al. [99] used the biology microplate method combined with the analysis of hydrogen peroxide and lignocellulolytic activity to determine the functional diversity of microbial communities in the fermentation materials of *A*. *subrufescens* cultivation. The results showed that lignocellulolytic activities, H_2_O_2_ production, and substrate transformation were not affected by the growth of *A*. *subrufescens* after 2 weeks. This method of measuring the metabolism of different carbon sources was used to evaluate the effect of spawn storage conditions on the substrate colonization of *A*. *subrufescens*. The results from the study reveal that the storage conditions of relatively low temperatures for up to 30 days did not affect the substrate colonization of this mushroom species [100]. This method exhibited higher sensitivity and simplicity than traditional plate culture. Moreover, it can effectively reduce cost and achieve higher degrees of efficiency.

##### Fingerprinting Based on Biochemical Components

The phospholipid fatty acid (PLFA) production profile can be used as a biomarker to analyze changes in microbial biomass and microbial community structure. PLFAs are important components of the cell membrane in vivo. Different groups of microorganisms can form different PLFAs through different biochemical pathways. The PLFA profile provides information on the microbial community structure, abundance of viable bacteria, and physiological status of microorganisms [101]. Vos et al. [102] quantified the biomass of living bacteria by analyzing PLFAs to study the specific changes in the microbial biomass in culture media during the colonization of *A*. *bisporus*. Song et al. [103] used PLFAs to compare the differences in microbial communities in the rice straw compost of *A*. *bisporus* cultivation prepared by different systems. The results indicate that the aeration-assisted compost had a higher microbial abundance, especially in terms of aerobic bacteria and Gram-positive bacteria, than conventional outdoor compost. In addition, aeration-assisted fermentation can effectively improve the degradation of rice straw material, accelerate the growth of mycelium, and increase the yield of fruiting bodies. The PLFA method can analyze the changes of all microorganisms in fermentation materials, including uncultured microorganisms with a large sample size. Even the metabolic characteristics of microbial community structures can be obtained via characteristic PLFA ratio analysis. A change in pH can affect the composition of microbial PLFAs [104]. Importantly, pH is dynamic in the fermentation process of culture materials, so there may be large errors in the changes in the microorganism community structure determined by PLFA analysis. Ellis and Ritz [105] developed high-throughput PLFA analysis technology through equipment optimization to solve the problem of the need for the consumption of many chemicals involved in the operation. With their method, extraction time can be reduced and the efficiency of the approach can be improved. In addition, PLFA analysis has been combined with the stable carbon isotope technique to indicate the response of microbial communities to environmental changes, and to facilitate the interaction between microbial communities [106]. Moreover, the other structural characteristic components within microbial cells are also widely used to classify and identify microorganisms in studied systems. Therefore, specific components are extracted, purified, and then classified and identified. Yu et al. [107] analyzed the quinone profile to monitor the microbial community during agricultural waste composting, and found that Actinobacteria was dominant during the thermophilic stage of the composting process.

#### 3.1.2. Culture-Independent Methods

Since many compost bacteria and fungi are not readily cultivable, studies that focused on single time points within the composting process have provided a limited overview. More recent DNA sequencing studies have provided evidence on the presence of other microbes in compost and their function, with a broad range of phyla involved in the microbial community during mushroom growth. DNA sequencing methods can detect almost all existing microorganisms in systems without the need to culture microorganisms. Understanding the microbiology of substrates provides researchers with necessary information that could be applied to design consortia of bacteria and fungi. These could then be used in bioaugmentation to optimize substrate quality, and to identify and validate biomarkers that could be used to assess the quality of compost before the start of cropping. More detailed studies are also required to explore the relationship between microbial activity and diversity in the substrate, compost, casing, and during cropping. Several methods and techniques have been applied in this area of study, e.g., denaturing gradient gel electrophoresis (DGGE), terminated-restriction fragment length polymorphism (T-RFLP), amplified rDNA restriction fragment analysis (ARDRA), and high-throughput sequencing. Most of these methods are based on polymerase chain reaction (PCR) and 16S or 18S rRNA as the “gold standard” in the study of evolutionary relationships. The results can reveal detailed information about the structure of the microbial community, including composition, abundance, and homogeneity. The resulting data can then be compared to the different species of microbes present in the samples.

##### Denaturing Gradient Gel Electrophoresis (DGGE)

The DGGE technique is based on the electrophoretic separation of double-stranded DNA produced by PCR in polyacrylamide gel containing chemical denaturants (urea and formamide). The bands visible in DGGE gels represent the composition of microbial communities. The more bands visible, the more complex the ecosystem is. Székely et al. [12] analyzed microbial communities in the composting process of *A*. *bisporus* cultivation by using DGGE and T-RFLP. They revealed that cellulose-degrading consortia were related to *Pseudoxanthomonas*, *Thermobifida*, and *Thermomonospora*. Souza et al. [56] studied thermophilic fungal populations during phase II compost production for the cultivation of *A*. *subrufescens* through DGGE analysis. It was found that *Scytalidium thermophilum* was the primary species in the entire process. It was also reported for the first time that *Theromyces badanesis* was involved in the substrate fermentation for *A*. *subrufescens* production. Chen et al. [108] studied the bacterial community change during the phase II composting of *V*. *volvacea* through 16S rDNA-DGGE analysis. The highest diversity of bacteria was found at the high-temperature stage. The dominant microbes in the entire fermentation process were heat-resistant bacteria and fungi. Most of the bacteria were identified as Proteobacteria, Bacteroidetes, and Firmicutes. Wang et al. [58] used PLFA analysis and the DGGE technique to compare the microbial species in rice and wheat straw, and found that microbial communities in rice straw were more abundant than those in wheat straw. The DGGE technique can be used for the reproducible assessment of microbial community structures, and is suitable for the analysis of community structure composition. However, PCR-DGGE results are influenced by the DNA-amplification process, and different DNA polymerases have been found to influence the produced fingerprints in PCR-DGGE analysis [109]. To obtain the full-length 16S rRNA gene sequence for bacterial taxonomy, a two-step method of denaturing gradient gel electrophoresis (DGGE) was developed by Wang and He [110]. The dominant microbes in the key stage are the primary analysis objects in the next step. This method can be utilized for the taxonomic identification of minor populations of interest from single or multiple microbial consortia. DGGE technology has universal application value in microbial community research. With DGGE, the resolution of dominant species is higher than that of non-dominant species, which has led to the long-term use of these methods in the research of fermentation materials.

##### Amplified Ribosomal DNA Restriction Analysis (ARDRA)

ARDRA is based on the conservation of rDNA sequences in prokaryotic and eukaryotic organisms. The rDNA fragments are amplified by universal or specific primers and digested by restriction enzymes. The fragments are then separated on high-density agarose or polyacrylamide gels. The microbial diversity is subsequently analyzed with a restriction enzyme map [111]. The diversity and community succession of culturable mesophilic bacteria in the early and late fermentation stages of *A*. *bisporus* are ultimately revealed by isolation culture and ARDRA techniques. It has been emphasized that bacterial aggregates may be helpful to accelerate the degradation process during the fermentation of the medium [112,113]. The ARDRA technique has the ability to distinguish bacterial species, but it is relatively difficult to carry out quantitative analysis and comparative analysis between populations [114]. However, when a large number of isolates need to be analyzed, the ARDRA technique can save significant amounts of time and cost [115]. Moreover, the technique has certain advantages and high reliability in the rapid identification of common microorganism species [116].

##### Terminal Restriction Polymorphism (T-RFLP)

T-RFLP technology is a further developed form of technology adapted from ARDRA. With this approach, the 16S rDNA or 18S rDNA gene is amplified using fluorescent dye-labeled primers. Then, the amplified fragment is digested by one or more restriction enzymes with four base-pair recognition sites. The restriction fragment is then detected by electrophoresis separation. The difference in the size of the terminal restriction fragments reflects the difference in 16S or 18S rDNA gene sequences (i.e., sequence polymorphism). Consequently, it is possible to analyze the phylogeny of different microbial populations and to evaluate the dynamics of microbial community structures [117]. Vajna et al. [118] used the T-RFLP technique to analyze bacterial succession in oyster mushroom (*Pleurotus* spp.) substrate preparation, and they identified and quantified the dominant flora in different stages. *Pseudomonas* and *Sphingomonas* were prevalent at the initial stage. *Bacillus*, *Geobacillus*, *Pseudoxanthomonas*, *Thermobispora*, and *Ureibacillus* were revealed at the end of partial composting, and finally, several genera of Actinobacteria, including *Bacillus*, *Thermobacillus*, *Thermus*, *Geobacillus*, and *Ureibacillus*, were found to be dominant in the mature substrate. Accordingly, the proportion of uncultured bacteria increased during the entire process. Székely et al. [12] used DGGE and T-RFLP for the analysis of microbial changes in the summer and winter composting cycles of *A*. *bisporus* cultivation, and found that thermophilic bacteria appeared earlier in summer than in winter. Thermophilic actinomycetes, *Pseudoxanthomonas*, *Thermobifida*, and *Thermomonospora*, were dominant as supposedly cellulose-degrading consortia at high temperatures. T-RFLP analysis showed that the change in bacterial community structure was related to the content of water-soluble carbohydrates, and the stability of substrate fermentative products and their potential application in agriculture or horticulture were verified. This technique provides highly reproducible results when compared to the relatively popular DGGE technique, due to the use of automated sequencers, while allowing for the rapid estimation of microbial abundance [119]. In practical applications, due to the different standards of analysis maps and the selection of restriction enzymes, the analysis results easily produce errors, which may limit the application of this method to some extent.

##### High-Throughput Sequencing

Next-generation sequencing (NGS) associated with bioinformatics analysis has significantly improved the knowledge on microbial communities by providing large visualizations of complex microbial communities and their metabolic potential data. A biotype, or an operational taxonomic unit (OTU), is used instead of a species to describe and compare microbial populations and communities [120]. Target DNA is cut into small pieces and binds a single small DNA molecule to a solid surface, which is copied only once and characterized by signal detection for every copy. NGS works in conjunction with a high-resolution imaging system, resulting in a high level of output and a high resolution [121]. NGS techniques in tandem with 16S rRNA amplicon sequencing for bacterial diversity, and 18S rRNA amplicon sequencing for fungal diversity, are widely used to study structural communities. The species and proportion of microorganisms in a habitat can be determined and classified into specific genera or species using these amplified sequences.

Metagenome sequencing can directly reflect the biological functions of microorganisms [122]. Vieira and Pecchia [123] studied bacterial communities under different pasteurization conditions with 16S rRNA amplification sequencing. Their results indicate that the bacterial community structure did not significantly change under different pasteurization conditions, but *Bacillus* showed higher abundances at high pasteurization temperatures. The growth rate of *A*. *bisporus* mycelium on the compost pasteurized at high temperatures was obviously slower than that on the compost pasteurized at low temperatures. The authors of this study preliminarily discussed the specific effect of pasteurization temperature on production, and the function of microbial communities discovered during substrate preparation needs to be further explored. The use of metagenomics has established the key position of thermophilic actinomycetes in lignocellulose biodegradation and its industrial potential for degrading enzymes [124].

The application of NGS technology is currently becoming less expensive, and bioinformatics tools and software programming for data analysis have been significantly improved. Hence, NGS technology is an ideal alternative to other molecular biological techniques, such as T-RFLP, because it can still be considered a low-cost and reliable alternative technique when analyzing a large number of samples [125]. To more deeply understand the diversity of microbial communities and to improve the accuracy of results, metagenomics should be used to explore the functional diversity of microbial communities on degrading cultivation substrates. Studies of microorganism diversity during fermentation based on the classification of bacteria and fungi have been carried out at differing levels of exploration. NGS technology is critical in studying the key bacteria involved in the fermentation process, and in exploring and verifying the indicative factors that can be used to assess the fermentation quality of culture media.

##### Quantitative Polymerase Chain Reaction (qPCR)

In addition to the above-mentioned techniques, quantitative PCR (qPCR) has also been applied to the analysis of microbial community diversity and its dynamics. The method is based on a PCR technique that couples the amplification of a target DNA sequence with the quantification of the concentration of that DNA species in the reaction. For example, a putative relationship between the amount of diseased *A*. *bisporus* mushrooms and the type of symptom was detected using qPCR [126], and the qPCR analysis of bacterial 16S rRNA and fungal ITS gene copies was used to study their abundance and community structure in cucumber continuous cropping soil treated with spent mushroom (*F. velutipes*) substrate [127].

### 3.2. Microbial Community Influence on Mushroom Cultivation

Substrates can be prepared and used in their natural, sterilized, or pasteurized forms, as mentioned in the methods for mushroom cultivation. The sterilization technique used in the process should be carefully applied due to its effect on the microbial communities in the substrate. Low bacterial cell numbers in spawn samples have been observed due to the rigorous thermal reduction, caused by cooking seeds in boiling water followed by sterilization in an autoclave [128,129]. Sterilization cultivation leads to a lower contamination ratio in a system than pasteurization. During the sterilization process, almost all microbes are killed. However, there is always a chance of contamination with microbes through the water spray that is loaded into the system to control moisture content. Longley et al. [130] cultivated morel mushrooms (*Morchella rufobrunnea*) indoors in pasteurized composted substrates, and then they compared microbial communities between successful fruiting and non-fruiting conditions. *Bacillus* and *Paenibacillus* belonging to Firmicutes were found to be the dominant bacteria during composting. *Pezizomycetes* were the dominant fungi in earlier states of the cultivation, and *Sordariomycetes* dominated in the cropping cycle. *Gilmaniella* was dominant in the successful fruiting system, while *Cephalotrichum* was dominant in the non-successful fruiting system. Understanding the microbial components of mushroom substrates could lead to their use as markers for suitable substrates and bioinoculants of particular taxa that have promise in promoting mushroom growth and yield [2]. Three groups of microbial communities should be considered during the mushroom cultivation cycle, including (1) microbial communities related the quality and maturing of the substrate, (2) microbial communities related to mycelium colonization on the maturing substrate, and (3) microbial communities related in fruitification, which may act separately or have interactive effects among them. Most reported microbial community changes have been related to physiochemical and metabolomics changes in the substrate’s preparation or composting. However, measurements of the property changes in mushroom-growing substrates are mostly not performed. It is probable that changes occur during microbial succession in substrates during mushroom growing, which could be researched in the future. The associations of microbial diversity with functions during substrate preparation could be different from those during vegetative mycelial growth or the development of fruiting bodies. Thus, the substrate used during fructification appears to be key in facilitating the necessary environmental conditions needed for fructification, such as physical and biological parameters, rather than providing nutrition to the developing fruiting bodies. The highlighted knowledge needed for the further exploration of microbial communities that facilitate or inhibit fructification will inform mushroom cultivation practices and enable successful large-scale production, as has been realized for individual cultivated mushroom species.

The study of microbial community dynamics is mostly conducted during substrate preparation for mushroom cultivation, and a few studies have been focused on the entire cultivation cycle. The physiochemical changes that occur in mushroom-growing substrates throughout cultivation have been studied [130], but those that occur in microbial communities have not been. Zhang et al. [131] revealed that the changes in physiochemical properties during *Ganoderma lucidum* growing may correlate with changes in microbial communities due to the creation of different niches that can be exploited by specific taxa. The role of the fermentation process on cultivation substrates involves the utilization of the metabolic activity of microbes that occur in either the raw material itself or in the air. The *Agaricus bisporus* cultivation cycle is mostly used as a model because its cultivation system heavily relies on ecological relationships with a broad range of microorganisms present in compost and casing. Several studies have focused on microbial communities and bacterial diversity at different phase of compost in *Agaricus* spp. [3,132]. Microbial communities during substrate preparation are important in the design of management strategies to optimize further cultivation systems. The physicochemical and biological properties of the fermentation media are changed during fermentation. Consequently, the fermented substrate is degraded and transformed to a selective substrate that is suitable for the growth of mushroom mycelia. These types of microorganisms constantly change during the fermentation process to adapt to dynamic changes in nutrient supply and environmental conditions. Therefore, the appearance or quantity of some microbes can reflect the degree of decomposition of cultures [133]. Physicochemical factors, such as temperature, oxygen, moisture, composting materials, and the C/N ratio, are known to affect microbial community changes in cultivation systems and in the development of mushrooms in substrates [134], or vice versa. A complex microbial community can affect composting structure changes, including temperature, pH, aeration, water content, and organic matter [135]. The analysis and study of microbial diversity and metabolism in fermented substrates will provide technical support for effective culture material preparation, and promote industrial mushroom cultivation.

Bacteria and fungi serve major roles in processing raw materials, minimizing fungal competitors, and inducing fructification [3,123,136]. Multiple bacteria and fungi play important roles in the population differentiation, environmental suitability, infection mechanism, and quality of *Ophiocordyceps sinensis* [16,137,138,139]. Proteobacteria, Firmicutes, Verrucomicrobia, and Deltaproteobacteria are the most relatively abundant bacteria detected during the life cycle of *O*. *highlandensis* [17]. Zhang et al. [16] observed microbial communities in the soil that have been associated with naturally occurring *C*. *militaris*. Acidobacteria, Actinobacteria, Bacteroidetes, and Proteobacteria were found to be enriched in the bacterial phyla, and *Ascomycota* was found to be the main fungal phylum. Genes related to metabolism were found to be more abundant in the soil bacteria, while membrane transport genes were found to be more abundant in the endophytic bacteria of *C*. *militaris*. Kong et al. [15] reported on the metabolic function of the microbial community in corncob composting systems. Aerobic and mesophilic microbial communities degrade organic matter during the mesophilic stage (T1, Day 2) of the composting process, and their metabolism can drive pH changes and increase the temperature to above 50 °C. Changing environmental conditions were found to promote the growth of a thermotolerant microbial community, and the temperature increased to 60–80 °C during the thermophilic stage (T2, Day 4). Bai et al. [140] observed that the different composting stages reflected differences in the bacterial community structure, and increased the abundance of genes related to the bacterial secretion system. The compost temperature was found to rapidly rise to 80 °C due to microbial activity during a period of aerobic thermophilic composting (phase I) [3,82,91]. Differing compositions of compost can result in differently dynamic dominant microbial communities. Bacterial communities may be involved in various interactions with mushrooms, such as atmospheric nitrogen control and triggering/inhibiting fruit body formation, inhibiting pathogens, and providing growth hormones [141]. The exploration of such bacteria and the identification of bacterial antagonists against various pathogens may be useful for improving cultivated mushroom yields [142,143]. The dominant bacterial species that have been detected in mushroom cultivation substrates are provided in Table 2. Metagenomic studies have shown that Actinobacteria and Firmicutes are the most dominant bacterial phyla present during fermentation, and both are regarded as highly active glucose fermenters [144,145].

The phyla Actinobacteria, Bacteroidetes, Firmicutes, Proteobacteria, and Planctomycetes were found to be primarily dominant in the *A*. *bisporus* cultivation cycle. Proteobacteria was detected throughout the cropping process, and the genus *Pseudoxanthomonas* was dominant during the composting process, but decreased during the cropping process. The phylum Thermodesulfobacteria was found to be highly abundant in the early stages of the cropping process [92]. Vieira and Pecchia [9] reported that Firmicutes and Actinobacteria, the dominant bacteria, appeared to have specificity distribution patterns in compost and casing soil, while Bacteroidetes appeared to have specificity distribution patterns in mushroom caps. Proteobacteria were found to be more uniformly distributed in all phases of the mushroom cultivation cycle. Community diversity tends to be higher in compost than in casing and mushroom samples. Firmicutes, a common fermenting group of bacteria, were found to become the dominant phylum at the thermophilic stage [146], and Proteobacteria re-emerged as the dominant phylum at the late stage of composting. This bacterial group has the ability to form heat-resistant endospores during the thermophilic phase [15,147]. Actinobacteria and Firmicutes are the dominant phyla in compost [148,149], and changes in community dynamics may be caused by different composting materials and systems. Shifting microbial community patterns were reported by Qiu et al. [150], who found that Actinobacteria and Bacteroidetes were reduced at the thermophilic stage but increased during the late stage of composting.

**Table 2 biology-11-00569-t002:** Dominant bacterial communities in different types of substrates.

Substrate Types	Dominant Bacteria		Properties Related to the Cultivation	Method of Analysis	Mushroom	References
	Phylum	Class/Order/Genus				
Wheat straw-based compost	Proteobacteria	*Pseudoxanthomonas*	Cellulose-degrading consortium	DGGE and T-RFLP analysis	*A. bisporus*	[12]
	Actinobacteria	*Thermobifida Thermomonospora*				
Corncob compost (Early stage)	Firmicutes	*Carnobacterium*	-	Metagenomic sequencing	*P*. *ostreatus*	[15]
	Proteobacteria	*Pseudomonas Stenotrophomonas*				
	Bacteroidetes	*Sphingobacterium*				
	Actinobacteria	*Glutamicibacter*				
Corncob compost (Thermophilic stage)	Firmicutes	*Aerococcus*	-			
		*Bacillus*				
		*Desemzia*				
		*Lysinibacillus*				
		*Enterococcus*				
	Proteobacteria	*Acinetobacter*				
	Actinobacteria	*Corynebacterium*				
Naturally occurring soil and mushroom	Acidobacteria	-	-	DNA sequencing	*C. militaris*	[16]
	Actinobacteria					
	Bacteroidetes Proteobacteria					
Soil in fruiting body	Firmicutes	-	-	16 rRNA amplicons, Illumina MiSeq sequencing	*O. highlandensis*	[17]
	Verrucomicrobia					
	Deltaproteobacteria					
	Proteobacteria					
Peach sawdust-based compost	Firmicutes	-	-	Metagenomic 16S rRNA sequencing	Oyster mushroom	[80]
	Actinobacteria					
	Proteobacteria					
Compost	Thermodesulfobacteria	*Thermodesulfobacterium*	Sulfur-reducing properties	DNA and cDNA sequencing	*A. bisporus*	[92]
	Proteobacteria	*Pseudoxanthomonas*				
	Actinobacteria	-				
	Firmicutes	-				
Natural composting samples (Mesophilic stage)	Actinobacteria	Actinomycetales Bacillales Clostridiales	-	DNA sequence with Roche/454 technology	-	[135]
	Firmicutes	*Rhodococcus* *Lactobacillus Thermobifida*	-			
	Actinobacteria	*Amycolatopsis*	Potential for lignin degradation			
Maize straw compost	Firmicutes	*Sporosarcina* *Bacillus* *Staphylococcus*	-	Illumina MiSeq sequencing	-	[151]
	Proteobacteria	*Pseudomonas* *Ochrobactrum*	-			
	Bacteroidetes	-	-			
	Actinobacteria	*Cellulosimicrobium*	Produce lignocellulose hydrolytic enzymes			
Sugarcane processing	Firmicutes	Bacillales Lactobacillales Clostridiales	Fermentation	PhyloChip microarray	-	[152]
	Proteobacteria	-	-			
	Bacteroidetes	-	-			
Wood chips and sawdust compost	Actinobacteria	*Brevibacterium*	Degrade cellulose	Real-time PCR and DGGE	-	[153]
		Micrococcineae	Degrade lignocellulosic materials			
		*Cellulomonas*	Produce cellulases and hemicellulases			
Composting from waste management system	Actinobacteria	-	-	DNA sequencing	-	[154]
	Bacteroidetes	-	-			
	Firmicutes	*Bacillus*	Produce proteases			
		*Clostridium*	Degrade cellulose and lignin			
		*Lactobacillus*	Related to low pH			
		*Thermoactinomyces*	-			
	Proteobacteria	*Acetobacter*	Related to low pH			
	Deinococcus-Thermus	-	-			

(-) = not determined in the reference.

The role of bacterial activities in the bio-transformation of substrates has been studied. *Thermus* species in the phylum Deinococcus-Thermus are typically thermotolerant aerobes that possess sulfur- and arsenic-reducing properties, and they are known to be sources of several enzymes with commercial applications at high temperatures [155]. The genus *Thermodesulfobacterium* of the phylum Thermodesulfobacteria was found to be sporadically active until the end of the early stages of mushroom cropping [92]. Actinobacteria are understood to play an important role in the biodegradation of cellulose and lignin [151]; however, this group has a significantly positive correlation with laccase and protease activities. Actinomycetales are abundant in thermophilic stages, and can degrade lignin and cellulose, which are recognized as complex and less complex carbon sources, respectively. This bacterial group can tolerate high pH and temperature values. *Rhodococcus* sp., *Thermobifida fusca*, and *Amycolatopsis* sp. have the potential for lignin degradation in compost with lignocellulosic biomass [135]. Actinobacteria, of the genera *Thermobifida*, and *Thermomonospora* were detected throughout the cropping process of *A*. *bisporus*. *Glutamicibacter* was predominantly observed in the early stages of the composting process. *Corynebacterium* exhibited the highest degree of abundance during the thermophilic stage of the composting process, and *Thermobifida* and *Mycobacterium* became the dominant genera at the late stage of the composting process. Additionally, *Cellulomonas* can degrade lignocellulose under neutral and alkaline conditions by producing endo-glucanase and exo-glucanase. This group of bacteria was observed to increase in the late period of composting [151,153]. *Thermobifida* was reported to be important in cellulose degradation [61]. The bacterial genus *Arthrobacter* has been reported to be present during the initial stages of the fruiting bodies of ground basidiomycetes.

Firmicutes have also been shown as the dominant phylum during the processing of complex carbohydrate sources such as sugarcane [152]. Bacillales tolerate high pH and temperature values, and degrade cellulose and solubilize lignin. Clostridiales, whether anerobic or microaerophilic species (as well as *Clostridium*), play important roles in cellulose degradation in composting [135]. *Lactobacillus* was found to be highly abundant at the start of the composting process [154]. *Bacillus* is dominant in lignocellulosic composting systems, and can contribute to waste degradation during the composting process due to its thermotolerance properties [156,157]. The presence of *Lactobacillus* has been correlated with low pH values (4.7–5.9) and mesophilic temperatures. However, some forms of them have been observed at pH 7 under thermophilic conditions [158]. *Carnobacterium* has been predominantly observed in the early stages of the composting process. *Aerococcus*, *Bacillus*, *Desemzia*, *Enterococcus,* and *Lysinibacillus* were found to exhibit the highest degrees of abundance during the thermophilic stage of the composting process. Proteobacteria were shown to be distributed in all microenvironments during the cultivation process of *A*. *bisporus* (composing, compost colonization, and casing colonization) [9]. *Acinetobacter* exhibited the highest degree of abundance during the thermophilic stage of the composting process, and *Pseudoxanthomonas* and *Luteimonas* became the dominant genera at the late stage of the composting process [151,153]. *Bradyrhizobium* sp. has been reported to be involved in N_2_ fixation, with a potential role in fungal growth or nutrition during the ascocarp development of *T*. *magnatum* [159]. Among the several members of Proteobacteria, Pseudomonadaceae were shown to be the most abundant in the casing and mushroom caps [9]. Bacteroidetes were found to be abundant in mushroom caps [9]; among these, *Sphingobacterium* was predominantly observed in the early stages of the composting process.

Complexes of microbial communities form during the fermentation process of substrate preparation. Actinobacteria and Firmicutes are thermotolerant genera that exhibit sulfur-reducing characteristics. Accordingly, they have been found to be highly active up to the first flush of *A*. *bisporus* cropping. It is possible that the consortia of Firmicutes and Actinobacteria present in the cropping compost substantially contribute to the bacterial fermentation of complex carbohydrates [92]. Wei et al. [151] revealed that the bacterial communities in corncob composting differed at different stages of maize straw composting; *Staphylococcus* in Firmicute, *Cellulosimicrobium* in Actinobacteria, and *Ochrobactrum* in Proteobacteria possibly participated in the transformation of humic acid. Notably, the consortium of Proteobacteria (*Pseudoxanthomonas*) and Actinobacteria (*Thermobifida* and *Thermomonospora*) were shown to be related to cellulose degradation in wheat straw compost [12]. A major complex microbial community associated with the mesophilic stage of the fermentation process is Lactobacillales in Firmicutes, and the complex microbial communities associated with the thermophilic stage are Actinomycetales in Actinobacteria, Bacillales, and Clostridiales in Firmicutes [135].

The role of bacterial activities in defensive responses to pathogens or the promotion of mycelia and mushroom growing has been studied. Aydoğdu et al. [160] found that *B*. *subtilis*, *Stenotrophomonas maltophilia*, *Pseudomonas rhodesiae*, and *B*. *amyloliquefacie* inhibited the growth of the green mold *Trichoderma aggressivum*, as well as *A*. *bisporus,* in vitro. Cho et al. [143] isolated fluorescent *Pseudomonas* spp. from the mycelial surface of *P*. *ostreatus* cultivation, and determined that the strains had positive effects on promoting basidiome formation and the growth of *P*. *ostreatus*. Proteobacteria and *Ps. putida* were found to stimulate mushroom fruiting body formation [161]. Bacillales members exhibited potential as a key group of important bacteria during phase I of composting, which has been found to be related to the breakdown of raw materials and the creation of a selective substrate for *A*. *bisporus* growth [123]. It seems likely that the *Agaricus* species may require an interacting consortium of both bacteria and fungi for effective lignocellulose breakdown, though further studies are required to confirm the relevance of this finding. The bioinoculant Actinobacteria and *Glutamicibacter arilaitensis* involved in cultivation have the potential to improve the biological efficiency of the oyster mushroom *P*. *florida* [162]. Benucci and Bonito [163] reported that the phylum Proteobacteria was most abundant, followed by Actinobacteria and Firmicutes, in the tissue samples of truffle species. Moreover, *Bradyrhizobium*, *Polaromonas*, and *Rhizobium* were found to be the most abundant bacterial genera within the fruiting bodies of tubers. The effect of the relevant layers in the fermentation media or compost piles on microbial diversity would likely be one of the important factors that affects the efficiency of fermentation. *Thermopolyspora* and *Thermobifida,* belonging to the phylum Actinobacteria, were shown to be dominant in the 30 cm layer of the compost pile. *Thermobifida* and *Thermopolyspora* were dominant microorganisms in the 100 cm layer of lignocellulosic-based compost piles, especially in maize straw composts, and their abundance increased in conjunction with high oxygen concentrations.

Fungi have been shown to appear at the cooling and maturation phases at the end of the composting process. Examples of fungal diversity detected in different substrate fermentations or mushroom cultivations are shown in Table 3. *Thermomyces lanuginosus*, a thermophilic fungus, has been reported to produce cellulase and hemicellulose enzymes. *Mycothermus thermophilus*, *Talaromyces thermophilus*, and *Thermomyces lanuginosus* were found to be most abundant during the high-temperature composting of the *A*. *bisporus* substrate [9,135]. Zhang et al. [61] reported that maize straw composts were dominated by the phylum Ascomycota, particularly the genus *Thermomyces*. *Thermomyces lanuginosus* produced various hemicellulases and cellulases during the process of composting, and this fungal strain was always present in the compost. *Scytalidium thermophilum*, *Thermomyces lanuginosus*, and *Thermomyces ibadanensis* were identified as the most abundant species during phase II of composting [56]. Salar and Aneja [94] reported that Ascomycota, specifically the thermophilic fungi *Chaetomium thermophilum*, *Malbranchea sulfurea*, *Thermomyces lanuginosus*, and *Torula thermophila* (*Mycothermus thermophilus*), promoted the growth of *A*. *bisporus*.

## 4. Volatile Organic Compounds Involved in the Microbial Community and Mushroom Cultivation

The volatile organic compounds (VOCs) produced by some mushrooms are predominantly C8 compounds, some of which inhibit primordial formation. In the cultivation of *A*. *biosporus*, the mushroom was found to produce VOCs during its vegetative and reproductive phases. Importantly, 1-octen-3-ol has been shown to be produced by *A*. *bisporus* mycelium [3,81], and this compound controls the early differentiation of vegetative hyphae to multicellular knots [3,168]. Moreover, 1-octen-3-ol and 2-ethyl-1-hexanol were shown to inhibit the primordia formation of *A*. *bisporus* and to be removed by the bacterial community in the casing layer [81]. The spore germination and growth of the pathogenic fungi, *Lecanicillium fungicola*, *Mycogone perniciosa* (wet bubble disease), and *Trichoderma aggressivum* (green mold disease), were found to be inhibited by 1-octen-3-ol, but the compounds also inhibited fruiting body formation [3,169]. The emissions of H_2_S gas from spent mushroom compost were found in high quantities as a result of the use of sulfur-based nutrients during the production and conditioning of the compost substrate [92,170,171]. Ethylene produced by *Agaricus* hyphae was reported to inhibit mycelial growth and primordia formation [172].

However, these results seem to contrast with those of Kurtzman [173], who reported that primordial formation was unaffected or stimulated by adding exogeneous ethylene. Other researchers have reported on the utilization of VOCs produced from mushrooms by microbes in cultivation systems. *Pseudomonas putida* could reduce the levels of ethylene produced by *Agaricus,* and encourage the fruitification of this mushroom [3]. Some bacteria, such as the genus *Thermodesulfobacterium,* exhibit sulfur-reducing properties [92]. The nitrifying and nitrification properties of *Stenotrophomonas* were characterized by Lee and Lee [174], and the increase in the activity of this genus also elevates nitrate levels in compost [92]. Pauliuc and Botau [175] reported that the VOCs produced by the oyster mushroom *P*. *ostreatus* exhibited inhibitory effects on *Bacillus cereus* and *B*. *subtilis*. Researchers have struggled for decades to determine which VOCs produced by mushrooms are relevant for fruiting body formation. Additional studies on VOCs, microbial communities, mushroom mycelial growth, primordia formation, and fruitification are needed to obtain vital information about their effects.

## 5. Conclusions

Substrate types and the potential of substrate utilization affect the biological efficiency of mushroom production. Different types of substrate materials can be used for cultivation, but optimal conditions must be met. Composting or fermentation processes are optional methods that can enable researchers to increase the properties of cultivation substrates. The functions of microbes in mushroom production systems involve the efficiency of the production system, including enzyme production, volatile utilization, and the stabilization of the optimal conditions. The study of the diversity of the microbial communities and their potential functions that inhabit substrate preparation and mushroom growth will provide information that will be useful for determining the functions of microorganisms in mushroom cultivation. To better understand the diversity of microbial communities and increase the accuracy of the results, metagenomics should be used in future research to explore the effect of the functional diversity of microbial communities on degrading cultivation substrates. Advanced analytical techniques and NGS technology are needed to further improve the study of the key bacteria involved in the fermentation process, and to explore and verify the indicative factors that can be used to assess the fermentation quality of culture media. To date, NGS and metabolomics studies have mostly been used to identify detailed links between the microbes and biochemical profiles in substrate preparation systems. The guiding microbes or metabolites associated with mushroom cultivation can be monitored and managed to affect the production efficiency of mushrooms. Traditional isolation and culture-dependent methods have not become useless because they can be used in more specific characteristic studies. Multiple techniques can be combined in experiments to better reflect microbial community information in terms of substrate fermentation and cultivation. On the other hand, knowledge of multiple disciplines is necessary in order to actively discuss and analyze the problems associated with the application of various technologies in the development of more suitable and effective systems for mushroom cultivation.

## Figures and Tables

**Figure 1 biology-11-00569-f001:**
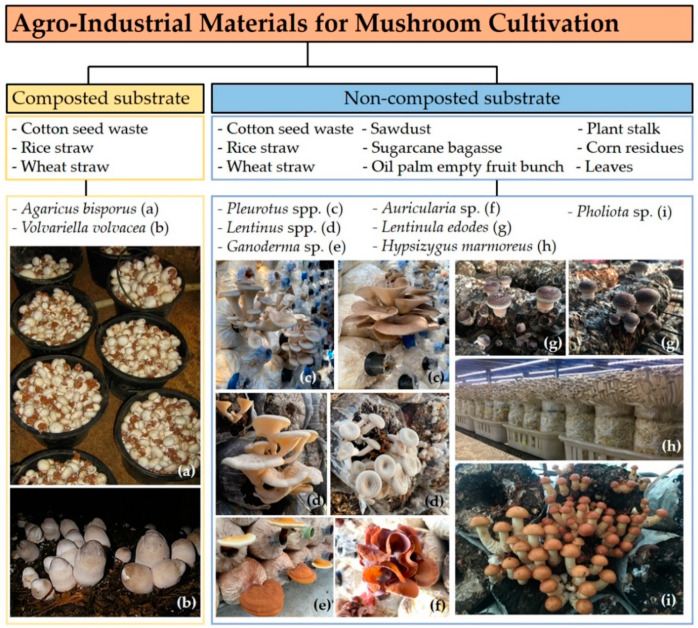
An illustration of the substrates used for mushroom production. Photo credit: Suwannarach, N.

**Figure 2 biology-11-00569-f002:**
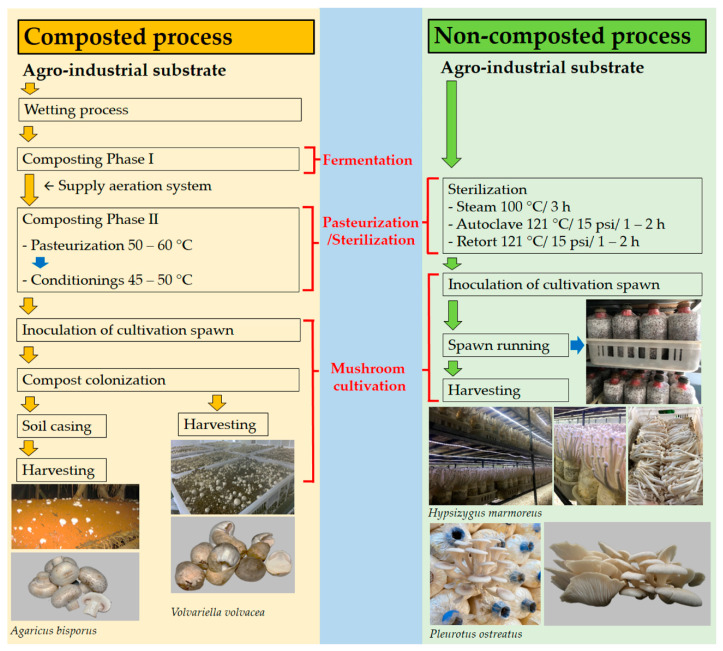
Overview of mushroom cultivation methods for composted and non-composted process. Photo credit: Suwannarach, N.

**Table 1 biology-11-00569-t001:** Examples of mushroom biological efficiency obtained on various agricultural materials.

Mushroom Cultivation	Agricultural Materials	Biological Efficiency (%)	References
*Hericium erinaceus*	Sawdust	50.3	[21]
	Wheat straw	43.5	
	Wheat straw	19.5	[22]
	Oak sawdust	37.3	
	Poplar sawdust	32.4	
	Common vetch straw	28.2	
*Volvariella volvacea*	Paddy straw	10.2–14.9	[23]
	Banana leaves	15.2	[24]
	Oil palm empty fruit bunch	3.6–6.5	[25]
*Ganoderma lucidum*	Oat straw	2.3	[26]
	Sawdust (*Swietenia mahagoni*)	4.3–7.6	[27]
	Sawdust (*Dipterocarpur turbinatus*)	3.6–6.8	
	Sawdust (*Tectona grandis*)	0.0	
*Auricularia polytricha*	Sawdust	90.0	[28]
	Sawdust	113.6	[29]
	Sawdust supplement with oil palm frond	184.8	
	Sawdust supplement with empty fruit bunch	195.6	
*Lentinula edodes*	Rice straw	36.1–49.7	[30]
	Rice straw	48.7	[31]
	Wheat straw	66.0	
	Barley straw	64.1–88.6	
	Sugarcane bagasse	130.2–133.4	[32]
	Sugarcane leaves	82.7–97.8	
	Bracts of pineapple crown	37.5–36.3	
*Pleurotus eryngii*	Ramie stalk	51.0	[33]
	Kenaf stalk	52.4	
	Bulrush stalk	36.8	
	Cotton seed hull	45.2	
	Wheat straw	48.2	[34]
	Rice straw	45.9	
	Corn cobs	51.8	
	Sugarcane bagasse	41.3	
	Sawdust	35.5	
*Pleurotus ostreatus*	Corn cob maize residues	14.0	[35]
	Composted sawdust	60.1	
	Beech sawdust	33.5	[36]
	Non-composted sawdust	4.3	[37]
	Composted sawdust	61.0	
	Composted sawdust	107.3	[38]
	Wheat straw	52.6	[36]
	Rice straw	50.6	[37]
	Banana leaves	37.1	
	Non-composted corncob	66.8	[39]
	Sawdust	46.4	
	Sugarcane bagasse	65.6	
	Wheat straw	105.0	[40]
	Wheat straw with spent coffee grounds	101.7	
*Pleurotus cystidiosus*	Non-composted corncob	50.1	[39]
	Sawdust	36.2	
	Sugarcane bagasse	49.5	
*Agrocybe cylindracea*	Beech sawdust	38.3	[36]
	Wheat straw	61.4	
	Wheat straw	23.0–36.0	[41]
*Flammulina velutipes*	Rubber tree sawdust and rice straw (1:1)	123.9	[42]
*Lentinus sajor*-*caju*	Wheat straw	74.9	[43]
	Rice straw	78.3	
	Soya stalk	83.0	
	Sunflower stalk	63.1	
*Agaricus bisporus*	Composted wheat straw	47.2–100.3	[44,45]
	Composted oat straw	47.2–52.9	[46]
*Agaricus subrufescens*	Composted wheat straw	6.6–53.7	[44,47]
*Agaricus blazei* (*A. subrufescens*)	One year-fermented horse manure bedding compost	62.1	[48,49]
	One year-fermented horse manure bedding compost with sawdust	24.9–27.7	
	One year-fermented horse manure bedding compost with corncob	20.0–25.3	
	One year-fermented horse manure bedding compost with woodchips	22.6–53.1	

**Table 3 biology-11-00569-t003:** Dominant fungal community in different types of substrates.

Substrate Types	Dominant Fungi			Properties Related to the Cultivation	Method of Use	Mushrooms	References
	Phylum	Class/Genus	Species				
Soil	*Ascomycota*	-	-	-	High-throughput sequencing of ITS gene	*C. militaris*	[16]
Composting (phase II)	*Ascomycota*	*Scytalidium* *Thermomyces*	*S. thermophilum* *T. lanuginosus* *T. badanensis*	-	PCR and DGGE	*A. subrufescens*	[56]
Peach sawdust-b ased composting	*Ascomycota*	Eurotiomycetes	-	Lignocellulosic degradation	Metagenomic ITS sequencing	-	[80]
		Sordariomycetes	-	Lignocellulosic degradation			
Compost (wheat straw)	*Ascomycota*	*Chaetomium* *Malbranchea* *Thermomyces* *Torula*	*C. thermophile* *M. sulfurea* *T. lanuginosus* *T. thermophila*	- - - -	Culture-dependent method	*A*. *bisporus*	[93]
Compost	*Ascomycota*	*Mycothermus*	Thermophilic fungi *My. thermophilus*	Produce lignocellulolytic enzymes	DNA recovery (Amplicon sequencing)	*A. bisporus*	[164,165]
		*Thermomyces*	*T. thermophilus*	Produce hemicellulase			[166]
		*Thermomyces*	*T. lanuginosus*	Produce xylanase			[167]

(-) = not determine in the reference.

## Data Availability

Data sharing not applicable.

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
