# Peer review of "Impact of Cultivation Substrate and Microbial Community on Improving Mushroom Productivity: A Review"

_biology, 2022, doi:10.3390/biology11040569_

Round 1
Reviewer 1 Report
A comment on lignin. (Line 162). I diagree that lignin is a source of Nitrogen that could be rendered available for the nutrition of the mushrooms. Lignin in degraded into phenolic compounds that contribute to the formation of N-rich humus complex by combinations with other sources of organic N. You should change your sentences (line 162-164)
A comment on composting: lines 285-287. There is a difference. Composting could improve the yield for Pleurotus but is necessary fo Agaricus which cannot be cultivated on raw ingredients. and the kind of composting is different.
A comment on 2.3. It is still confusing. We move from one enyme to the other and one species to the other without transitions or physical marks of changes, and the reader is rapidly borred because he/she canot find any story telling of essential points.
A comment on metagenomics. with the sentence line 552-554, I waited for data on true metagenomics (sequencing of all the DNA an recnostruction of sequences from all the microorganisms and not on amplicon sequencing (or metabarcoding) that is explaine after. Please be clear on the definitions ans what was really done in the cited papers.
Table 1 is not exhaustive. You should change the title. i.e. 'Exemples of mushroom BE obtained on various agricultural materials'. You should group the two lines on A. subrufescens (ref 44 and 48), even if in 48 the author used the false name A. blasei.
The added text should be checked for the English language. There are some grammar problems ans sentences are difficult to understand. Few exemples:
line 67-6. I do not understand. Line 71 : may be add 'produced' between 'substrate' and 'through'.
line 148-149. It should be : 'Bellettini et al. [54] summaried actors affecting the production of Pleurotus....'
line 194: I do not understand the end of the sentence, which relationships?
Line 208: are illustrated, not is illustrating
Line 254 : beech, not breech
Line 403: metabolome means all the metabolites produced by A.s. That is not what is in the paper.
I stop here to highlight the difficulties in english lungage here, but you should ckeck all the text added in this version of the manuscript.
Reviewer 2 Report
Impact of Cultivation Substrate and Microbial Community to Improve Mushroom Productivity: A Review
The structure of several paragraphs has been improved and the main object of them appears more clearly. Some of them have been shared.
Nevertheless, one paragraphs (L 172-194) is still badly structured and English must be improved before reviewing process for L 594 to 643:
L 172-194 The first sentence doesn’t correspond to the object of the paragraph, object which is not obvious. “The biological properties of the composts” have been specified in the first sentence or de preceding paragraph as “population and activity of microbes”. However, the paragraph (L172-194) doesn’t evoke population of microbes, deals mainly with chemical properties of the compost and with agricultural materials used as substrates. The second and third sentences, dealing with thermophilic stage are the only one corresponding to activity of microbes in the compost. If you talk of the role of mycelium network of M. importuna, you should clearly distinguish (yet in the first sentence of the paragraph) the compost before or at the date of inoculation and the modification of this compost by the fungus during the vegetative phase of the mushroom cultivation. The importance of these biological properties for the induction of fruiting doesn’t appear at all in the paragraph! Then you should clarify the object of the paragraph or supress it.
L 148 suggestion: replace “summary” by “summerize”
L 151 suggestion: … and humidity that affect the production of Pleurotus sp.
L 162 Lignin doesn’t provide nitrogen as there is no nitrogen in lignin molecule, but degradation products of lignin have affinity for peptides and amino acids and generate nitrogen-rich complex.
L 207-208 suggestion: The cultivation methods used in mushroom production is illustrated in Figure 2.
L 339 suggestion : Aslani et al. [77] have been previously reported that
L 540 suggestion: the bioinformatics analysis
L 540 suggestion: Next-generation sequencing (NGS) associated with bioinformatics analysis
has greatly improved the knowledge on microbial community by …
L 594 – 643 English must be improved before reviewing process.
L 599 suggestion: Sterilization cultivation cause the lower contamination ratio in the system than …
L 600-602 English: During sterilization process, almost all microbes are killed. However, there is always a chance of contaminating with the microbes through the water spray that loads into the system to control moisture content.
L 606 English: The dominant fungi that occured in earlier stages of the cultivation were …
L 607-608 suggestion - English: Gilmaniella was dominant in the successful fruiting system, while Cephatrichum was dominant in non-successful fruiting system.
– L 643 Insufficient English to understand the text
L 693-694 suggestion: Vieira and Pecchia [9] reported that Firmicutes and Actinobacteria, the dominant bacteria, seem to have specific distribution patterns
L 868 What do you mean by “guiding of interest markers”?
Moreover, the sentence is too long.
Reviewer 3 Report
Authors made big improvements to the manuscript and are well thanked for that. They took into consideration the majority of the raised suggestions and recommendations. Despite the improvements made in terms of the manuscript’s language, it still needs minor modifications that can be easily dealt with by the authors. No major comments were addressed, only minor adjustments are needed overall the manuscript. The presented parts are significant and are interpreted appropriately. The raised conclusions and suggestions are justified. The review covers its topic which is well relevant and all used references by the authors are appropriate. The review is correctly designed and sounds technically. Further references were suggested as they match with all the raised and discussed points by the authors.
Briefly, based on the above and below detailed explanation, the manuscript needs minor adjustments and surely has a merit to be published in “Biology” journal once all suggestions and recommendations are fully addressed.
Abstract
1- Page 1, lines 25–27: “Moreover… cultivation”: Kindly avoid the first voice form of the sentence and adopt the impersonal form instead.
Introduction
1- Page 2, lines 67–68: “However… elucidated”: The sentence is badly written in standard English; accordingly, kindly reformulate it.
2- Page 2, line 69: Kindly adjust as follow: “involved”.
- Mushroom cultivation
1- 2.1. Sources and Composition of Substrates: Page 5, lines 129–131: Kindly adjust these sentences as follow: “The substrates used for effective cultivation of mushrooms are shown in Figure 1. Volvariella volvacea requires substrates with high cellulose and low lignin contents; it produces a variety of cellulase enzymes for cellulose degradation.”
2- 2.1. Sources and Composition of Substrates: Page 5, lines 148–149: Kindly adjust as follow: “summarized that factors…”
3- 2.1. Sources and Composition of Substrates: Page 5, line 160: Kindly adjust as follow: “of principal”.
4- 2.2. Methods for mushroom cultivation: Page 7, line 208: Kindly adjust the sentence as follow: “… production are shown in Figure 2”.
5- 2.2. Methods for mushroom cultivation: Page 7, line 209: Kindly adjust as follow: “without it”.
6- 2.2. Methods for mushroom cultivation, 2.2.1. Cultivation on Non-Composted Substrates: Page 8, line 227: Kindly adjust the sentence as follow: “were found the most suitable for cultivation… with BE…”
7- 2.2. Methods for mushroom cultivation, 2.2.1. Cultivation on Non-Composted Substrates: Page 8, line 230: Kindly adjust as follow: “ranged between”.
8- 2.2. Methods for mushroom cultivation, 2.2.1. Cultivation on Non-Composted Substrates: Page 8, lines 252–253: Kindly remove “production” before “yield”.
9- 2.3. Enzymes Involved in Substrate Utilization and Mushroom Growth: Page 9, lines 307–308: Kindly adjust as follow: “have been previously identified”.
10- 2.3. Enzymes Involved in Substrate Utilization and Mushroom Growth: Page 9, line 339: Kindly remove “has been”.
- Microbial community for mushroom cultivation
1- 3.1. Methods Used for Analysis of Microbial Communities, 3.1.2. Culture-Independent Methods, 3.1.2.4. High-Throughput Sequencing: Page 14, lines 548–549: Kindly adjust as follow: “to study the structural communities”.
2- 3.2. Microbial Community Influence on Mushroom Cultivation: Page 15, line 594: Kindly remove “form”.
3- 3.2. Microbial Community Influence on Mushroom Cultivation: Page 15, line 595: Kindly adjust as follow: “forms as mentioned”.
4- 3.2. Microbial Community Influence on Mushroom Cultivation: Page 15, lines 599–600: “Sterilization… method”: The sentence is badly written in standard English; accordingly, kindly reformulate it.
5- 3.2. Microbial Community Influence on Mushroom Cultivation: Page 15, lines 601–602: “However… content”: Same recommendation as in the previous comment.
6- 3.2. Microbial Community Influence on Mushroom Cultivation: Page 15, line 604: Kindly adjust as follow: “conditions”.
7- 3.2. Microbial Community Influence on Mushroom Cultivation: Page 15, lines 606–607: “The dominant… cycle”: The sentence is badly written in standard English; accordingly, kindly reformulate it.
8- 3.2. Microbial Community Influence on Mushroom Cultivation: Page 15, lines 607–608: Kindly adjust as follow: “was dominant”.
9- 3.2. Microbial Community Influence on Mushroom Cultivation: Page 15, lines 609–611: “Understanding… [2]”: The sentence is badly written in standard English; accordingly, kindly reformulate it.
10- 3.2. Microbial Community Influence on Mushroom Cultivation: Page 15, lines 611–612: “Three… including”: Same recommendation as in the previous comment.
11- 3.2. Microbial Community Influence on Mushroom Cultivation: Page 15, lines 613–615: Kindly adjust as follow: “related to”.
12- 3.2. Microbial Community Influence on Mushroom Cultivation: Page 15, lines 625–628: “The highlight… species”: The sentence is badly written in standard English; accordingly, kindly reformulate it.
13- 3.2. Microbial Community Influence on Mushroom Cultivation: Page 15, line 629: Kindly adjust as follow: “is mostly done during substrate preparation”.
14- 3.2. Microbial Community Influence on Mushroom Cultivation: Page 15, line 631: Kindly adjust as follow: “that occur”.
15- 3.2. Microbial Community Influence on Mushroom Cultivation: Page 16, lines 693–694: Kindly adjust as follow: “reported that the dominance of… appears to have…”
16- 3.2. Microbial Community Influence on Mushroom Cultivation: Page 16, line 696: Kindly adjust as follow: “in all phases”.
17- 3.2. Microbial Community Influence on Mushroom Cultivation: Page 17, line 707: Kindly adjust as follow: “have been studied”.
18- 3.2. Microbial Community Influence on Mushroom Cultivation: Page 18, line 769: Same recommendation as in the previous comment.
Round 2
Reviewer 2 Report
With this third correction, reviewers’ comments have been considered and the English has been much improved.
The article merits to be published in “Biology” journal with minor modifications proposed below:
L 182
Agaricus brasiliensis is a synonym of A. subrufescens Peck
https://www.gbif.org/species/7747622
Agaricus brunescens does not exist in nomenclature of fungi (see index fungorum and species fungorum: http://www.speciesfungorum.org). Moreover, this name is absent in the three references you cite (61-63).
Agaricus brunnescens Peck (with two n) exist in Species fungorum, but it is not cultivated. Latest specimen found was in 2005 (https://www.gbif.org/species/5243459) and no DNA sequence are published in NCBI database (https://www.ncbi.nlm.nih.gov/search/all/?term=Agaricus%20brunnescens)
Brown stains of Agaricus bisporus are sometimes called Agaricus brunnescens or Portobello:
« portobello (Agaricus brunnescens), variété brune du champignon de Paris (Agaricus bisporus) » (https://fr.wikipedia.org/wiki/Portobello).
Agaricus bisporus “When marketed in its mature state, the mushroom is brown with a cap measuring 10–15 centimetres (4–6 inches). This form is commonly sold under the names portobello”, … (https://en.wikipedia.org/wiki/Agaricus_bisporus)
Please indicate the valid name for these two fungi (A. subrufescens and Agaricus bisporus)
L 302 suggestion: Many of the of the enzymes
L 3017 suggestion: … but the enzyme concentration significantly increased …
L 560 3.1.2.5
L 586 Cephalotrichum
Table 1 I suggest to follow advices given by reviewer 1:
“Table 1 is not exhaustive. You should change the title. i.e. 'Exemples of mushroom BE obtained on various agricultural materials'.”
